# Is personalised prehabilitation feasible to implement for patients undergoing oncological treatment for lung cancer at a London teaching hospital? Protocol of a feasibility trial

Kelly Wade-Mcbane ![ORCID],[1] Alex King ![ORCID],[2] Catherine Urch ![ORCID],[2] Lina Johansson ![ORCID],[1] Mary Wells ![ORCID] [3]

[1]Department of Nutrition and Dietetics, Imperial College Healthcare NHS Trust, London, UK
[2]Department of Surgery, Cardiovascular and Cancer, Imperial College Healthcare NHS Trust, London, UK
[3]Department of Surgery and Oncology, Nursing Directorate, Imperial College Healthcare NHS Trust, London, UK

**Correspondence to**
Kelly Wade-Mcbane;
k.wade-mcbane@nhs.net

## ABSTRACT

**Introduction** There is significant potential to improve outcomes for patients with lung cancer in terms of quality of life and survival. There is some evidence that prehabilitation can help, but, to date, this has only been tested in surgical populations, despite 70%–80% of patients with lung cancer in the UK receiving non-surgical treatment. The physiological and psychological benefits of prehabilitation seen in surgical patients could be extrapolated to those receiving non-surgical treatment, particularly in such a poor prognosis group. With patients and healthcare professionals, we have co-designed a personalised and evidence-based prehabilitation programme. This draws on a conceptual framework that aligns with patient values and needs as well as functional goals. We aim to investigate whether this programme is feasible to implement and evaluate in clinical practice.

**Methods and analysis** An open-label, single-group feasibility study incorporating quantitative assessments, a qualitative free text questionnaire and reflective field notes. Thirty participants will be recruited over an eight-month period from a single London teaching hospital. All recruited participants will receive a personalised prehabilitation programme during their oncological treatment. This includes a one-hour face-to-face appointment prior to, at week three and at week six of their treatment regimen as well as a weekly telephone call. Interventions including nutrition, physical activity and psychological well-being are stratified according to a patient's priorities, level of readiness and expressed needs. The primary outcome will be feasibility of the personalised prehabilitation programme in clinical practice by investigating areas of uncertainty regarding patient recruitment, attrition, treatment fidelity, intervention adherence and acceptability of study outcome measures. Secondary outcomes will include quality of life, functional capacity and grip strength.

**Ethics and dissemination** Ethical approval has been obtained from the Health Research Authority (reference number 22/PR/0390). Results of this study will be disseminated through publication in peer-reviewed articles, presentations at scientific conferences and in collaboration with patient and public involvement representatives.

---

## STRENGTHS AND LIMITATIONS OF THIS STUDY

⇒ Feasibility study to assess whether a future definitive study can or should be done.
⇒ Inclusive research approach, tailoring interventions to a participant's capabilities.
⇒ Use of a stratified person-centred prehabilitation intervention co-designed with patient partners.
⇒ Risk of potential bias owing to the lead researcher also being the clinician delivering the interventions.

---

**Trial registration number** NCT05318807.

## INTRODUCTION

Multimodal prehabilitation programmes incorporating nutrition, physical activity and psychological well-being are increasingly being used in surgical cancer pathways. This is in order to optimise health outcomes and enhance recovery by reducing postoperative complications[1–5] and hospital length of stay[4 6–8] as well as improving health reported quality of life.[9–12] Few prehabilitation pathways exist for people who do not have surgery, despite the fact that 50%–60% of people with cancer in the UK are treated with primary, neoadjuvant or palliative chemotherapy and/or radiotherapy treatment.[13] A small number of studies have investigated the use of prehabilitation in the neoadjuvant setting and have shown benefit in terms of postsurgical outcomes,[14–18] but only one of these have been in lung cancer.[19]

Only a minority of people with lung cancer are eligible for surgery, with 70%–80% receiving non-surgical treatment within the UK[20] without any prehabilitation. UK deaths from lung cancer are higher than any other cancer and treatment outcomes are poor.[21] There is significant potential to improve

outcomes for people with lung cancer and there is some evidence that prehabilitation can help. However, this has only been tested in surgical populations. A small number of studies have shown that prehabilitation before lung cancer surgery is feasible and can improve functional capacity and reduce postoperative complications. However, most of these studies focused on the benefits of exercise prehabilitation as opposed to a multimodal approach.[19 22–30]

Prehabilitation during chemotherapy, radiotherapy or immunotherapy treatment for lung cancer poses significant challenges. Treatment regimens are often prolonged and people may experience a range of toxicities, which could limit their ability to engage in prehabilitation interventions. Personalisation to person-led values and needs is therefore important, as initiation and adherence to any intervention is determined by behavioural, psychological, physiological, environmental and social factors.[31–33] This is further supported by Bayly et al's research in thoracic lung cancer[34 35] which found that participants had varied needs, highlighting the importance of tailored support to optimise symptom control, treatment tolerance and independence.

We have co-designed a multimodal personalised prehabilitation programme with healthcare professionals and patients using a three-step participatory action research approach (see online supplemental material file 1).[36–38] The multimodal structure of the programme is informed by the Macmillan, Royal College of Anaesthetists and National Institute of Health Cancer and Nutrition Collaboration, Research Principles and Guidance for Prehabilitation within the Management and Support of People with Cancer.[39] This guidance incorporates nutrition, physical activity and psychological well-being using a tiered approach, tailored to whether patients need universal (applicable to anyone with cancer), targeted (applicable to those at risk of acute, chronic or latent effects of disease or treatment) or specialist (applicable to those with complex needs, severe impairment and/ or disability) intervention.[39] The first level of intervention—universal—comprises of written information and relevant signposting to support resources for all patients. If psychological, nutrition and/or physical activity needs are identified, patients then receive the appropriate intervention (targeted or specialist) according to their level of need and readiness.

The personalised prehabilitation programme uses the Adversity, Restoration and Compatibility (ARC) framework[32] to underpin a tailored and collaborative approach. The ARC framework provides a synthesised view of how people conceptualise the personal experience of living with and beyond cancer. Consistent with psychological adjustment theory,[40] it frames this experience as an ongoing process of learning about the evolving challenge (adversity), learning how to cope effectively (restoration) and adapting one's identity (compatibility), in parallel (see figure 1). Our programme provides a structure and process for personalising prehabilitation, which is person-centred rather than providing a purely logistical or psychometric approach. As recommend by Hickmann et al,[41] such an approach could be key to empower patients to maintain progress through the prolonged, more variable context of perioncological prehabilitation.

This proposed feasibility study aims to investigate whether a personalised prehabilitation programme codesigned with healthcare professionals and patients and informed by theory and evidence is feasible to implement and evaluate in clinical practice and could thereby support a future definitive study.

## METHODS AND ANALYSIS

Feasibility studies are particularly relevant to assess whether a future definitive study can or should be done.[42–44] Using the Orsmond and Cohn framework for feasibility studies,[45] this open-label, single-group study will use a mix of methods to understand the different aspects of feasibility. This will be through the inclusion of quantitative assessments, a qualitative free text questionnaire and reflective field notes to investigate areas of uncertainty regarding patient recruitment, attrition, treatment fidelity, intervention adherence and acceptability of study outcome measures. This is situated within the feasibility stage of the Medical Research Council (MRC) framework for developing and evaluating complex interventions.[46] It will help inform if the personalised prehabilitation programme can be implemented and evaluated in clinical practice and will provide data to estimate the parameters required to design a future definitive study. The Consolidated Standards of Reporting Trials (CONSORT) reporting guideline for pilot and feasibility studies[42] has been used for this protocol (see online supplemental material file 2).

### Eligibility criteria

Inclusion criteria: Adults (≥18 years) with a diagnosis of lung cancer who are due to commence chemotherapy, radiotherapy or immunotherapy treatment. This study includes all lung cancer types and stages. People with comorbidities and those on medication will be considered eligible. Interventions will be tailored to a participant's capabilities, thereby promoting inclusivity and optimisation of participant recruitment.

Exclusion criteria: Participants who are due to undergo lung cancer surgery, have had previous lung cancer surgery or unable to give informed consent will not be eligible for this study.

### Recruitment

Participants will be recruited from the lung cancer clinic at a single London teaching hospital via referral from the Lung Cancer Multi-Disciplinary Team. Recruitment will continue until 30 participants have been recruited or until eight months have elapsed. A pragmatic sample size of 30 participants over an eight-month recruitment period from November 2022 to June 2023 has been selected,

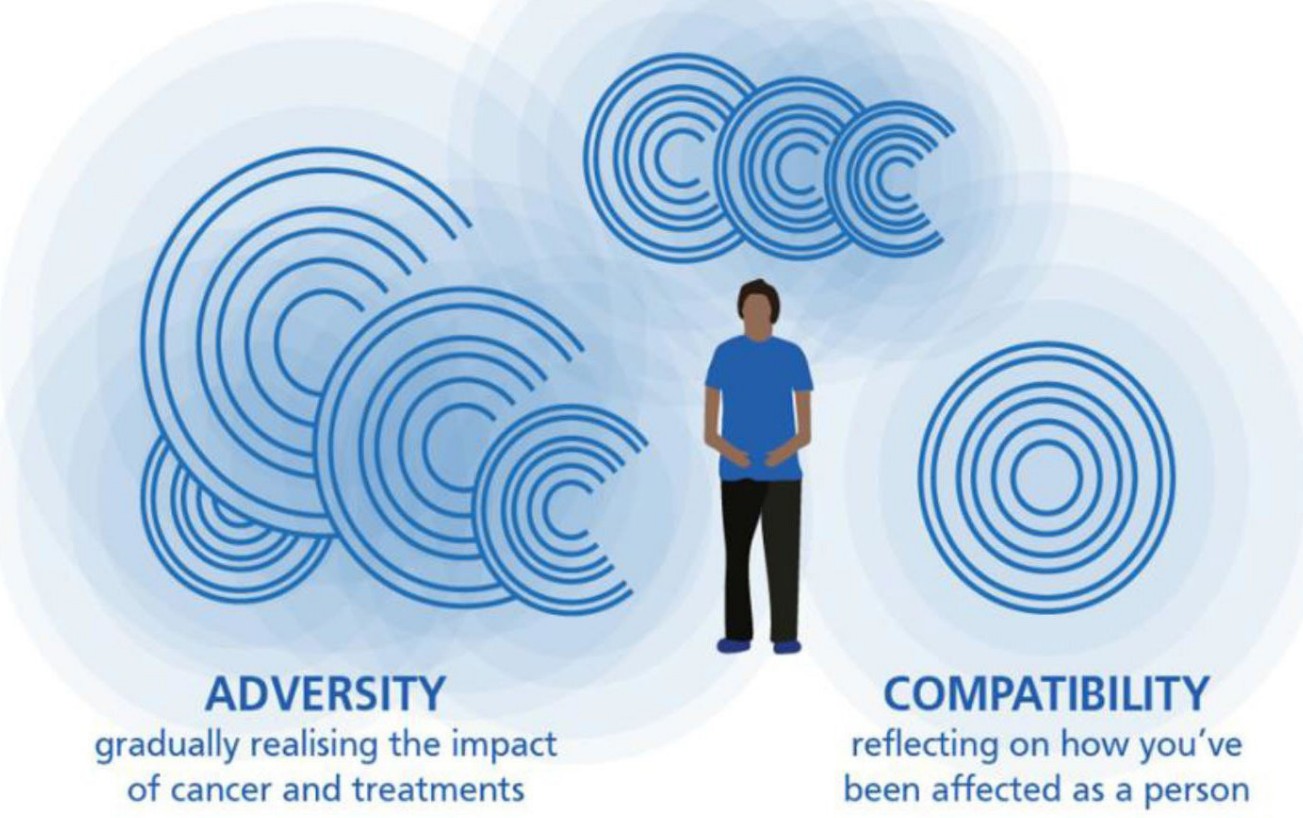

**RESTORATION**
gradually learning what you can do
to cope with all the new challenges

**ADVERSITY**
gradually realising the impact
of cancer and treatments

**COMPATIBILITY**
reflecting on how you've
been affected as a person

**Figure 1** Diagrammatic representation of the Adversity, Restoration and Compatibility (ARC) model of the experience of adjusting to living with and beyond cancer. The waves and curves reflect the fluidity and non-linear nature of the process, with the sizes reducing as one adjusts to living with and beyond cancer.

which is feasible within the time frame of this study. This is based on 144–240 new lung cancer diagnoses per year at the study hospital, with an assumption of a 50% uptake over an eight-month period. Furthermore, it is in accordance with published guidance on recommended sample sizes for feasibility studies.[47 48]

### The personalised prehabilitation programme

All participants will be required to attend three face-to-face appointments at the study hospital for study assessments, but the interventions will be delivered using a hybrid approach; either face-to-face at the study hospital or remotely via telephone or video conferencing depending on participant preference.

The three face-to-face sessions with the lead researcher will be prior to, at week three and at week six of a participant's oncological treatment regimen.

Following consent, participants will be guided through a set of questionnaires (as detailed below) to complete prior to their first face-to-face appointment. These questionnaires will be reviewed during the initial session, enabling the nutrition, physical activity and/or

psychological well-being interventions to be stratified according to a participant's priorities, level of readiness and expressed needs:

▶ Cancer Behaviour Inventory (CBI) brief form: to measure self-efficacy for coping with cancer. A score of less than 50% indicates a low level of readiness, whereas a score of 50% or greater indicates a high level of readiness.[49]

▶ Patient Activation Measure (PAM): to measure a person's knowledge, skills and confidence in managing their own health and well-being. A score of 1–2 indicates a low level of readiness, whereas a score of 3–4 indicates a high level of readiness.[50]

▶ Modified Godin leisure time exercise questionnaire: to measure activity. A score of 14 or greater demonstrates that a person is active, thereby indicating a low level of need for physical activity. Conversely, a score less than 14 demonstrates that a person is insufficiently active, thereby indicating a high level of need to optimise activity levels.[51]

- Patient Generated Subjective Global Assessment (PG-SGA) short form: to identify malnutrition risk. A score of less than 4 demonstrates no or minimal risk for malnutrition, thereby indicating a low level of need for nutritional support. Conversely, a score of 4 or greater demonstrates that a person is at risk of malnutrition or is malnourished, thereby indicating a high level of need for nutrition support.[52]
- Emotions Thermometer (ET): to evaluate levels of emotional distress. A score in any domain of less than 5 or a score of less than 4 in need for help demonstrates low levels of emotional distress, thereby indicating a low level of need. Conversely, a score of 5 or greater in any domain or a score of 4 or greater in need for help demonstrates high levels of emotional distress, thereby indicating a high level of need.[53]
- Patient Generated Index (PGI): to identify participant priorities.[54]

Participant needs will also be determined objectively using the six minute walk test (6MWT) to assess functional capacity and handgrip strength to assess upper body strength. Participants will undertake these functional assessments during the face-to-face sessions.

Questionnaire responses and outcomes from the functional measurements will inform the conversation with the participant about their priorities and enable interventions to be stratified according to a participant's level of readiness and need. If a subjective (questionnaire) or objective measurement (6MWT or handgrip strength) indicates a low or high level of readiness or need, the intervention will be tailored to the individual accordingly (see figure 2A–C).

The personalised prehabilitation programme uses a universal, targeted and specialist approach to prehabilitation interventions, in line with current guidance.[39] All participants will receive a universal intervention comprising of written information and relevant signposting to cancer-specific nutrition, physical activity and/or psychosocial well-being support resources. If a participant identifies a psychological, nutrition and/or physical activity need, they will then receive the appropriate intervention according to their level of need and readiness (see figure 2A–C). Targeted interventions will be delivered via the use of certified apps and/or specific group activities at Maggie's cancer centre (an independent charity that provides free cancer support, workshops and courses to help anyone affected by cancer). For example, exercise classes or nutritional educational sessions for participants undergoing treatment as well as the opportunity to attend a lung cancer support group. These group activities are run by cancer support workers or qualified instructors. Specialist interventions involve a 1:1 assessment with a registered healthcare professional such as a physiotherapist, dietitian and/or a clinical psychologist depending on a participant's needs. The objectives of specialist intervention with the physiotherapist and dietitian are to increase physical activity and optimise dietary intake, respectively, to minimise loss of body weight and muscle mass. Specialist intervention with the physiotherapist will include tailored advice regarding resistance and aerobic exercise. In comparison, specialist intervention with the dietitian will include a full individualised nutritional assessment, taking into account anthropometric, biochemical and clinical parameters as well as undertaking a comprehensive diet history and calculation of nutritional requirements to minimise further malnutrition.

Owing to the prolonged nature of chemotherapy, radiotherapy or immunotherapy treatment in contrast to a defined one-off target such as surgery, person-led values and needs are likely to be more variable within the context of peri-oncological prehabilitation. Throughout treatment, participants will be asked to keep a daily diary to record their symptoms, appetite, mobility and level of distress each day using a Likert scale. This will provide participants with a visual representation of their progress and help identify any adverse trends.

Participants will also receive a weekly telephone call from the lead researcher (a specialist oncology dietitian). Participants will be able to use the diary to highlight any days they found particularly good or difficult and have the opportunity to discuss these. Personalisation is central to this study and the language used during the face-to-face appointments and the weekly telephone calls has validation and compassion at its core to help build self-efficacy and sustain behaviour change.[55] The use of the PGI to understand personal priorities is fundamental to this personalised approach. A systematic review[56] demonstrated the large potential value of using the PGI by enabling people with cancer to participate in the decision-making process. This is key to empower people to take ownership of their own health and for the sustainability of long-term positive behaviour change and self-care.

A traffic light system with varying levels of acceptability has often been cited as a means of judging feasibility.[42 44 57–60] Following consensus among the research team, a priori feasibility endpoints were defined using red, amber, green (RAG) criteria as shown in table 1.

## Primary objective and outcome measure

The primary objective is to assess the feasibility of the personalised prehabilitation programme in clinical practice by gathering data on how accessible and relevant it is for patients with lung cancer undergoing chemotherapy, radiotherapy or immunotherapy treatment. This will be addressed using the research objectives and methods of assessment outlined in table 2.

Heterogeneity of participant characteristics is expected within the study. This will provide valuable insights into how different people experience the personalised prehabilitation programme.

## Secondary objectives and outcome measures

Functional benefits of prehabilitation can be seen in as little as 2 weeks.[39 61] The secondary objectives will

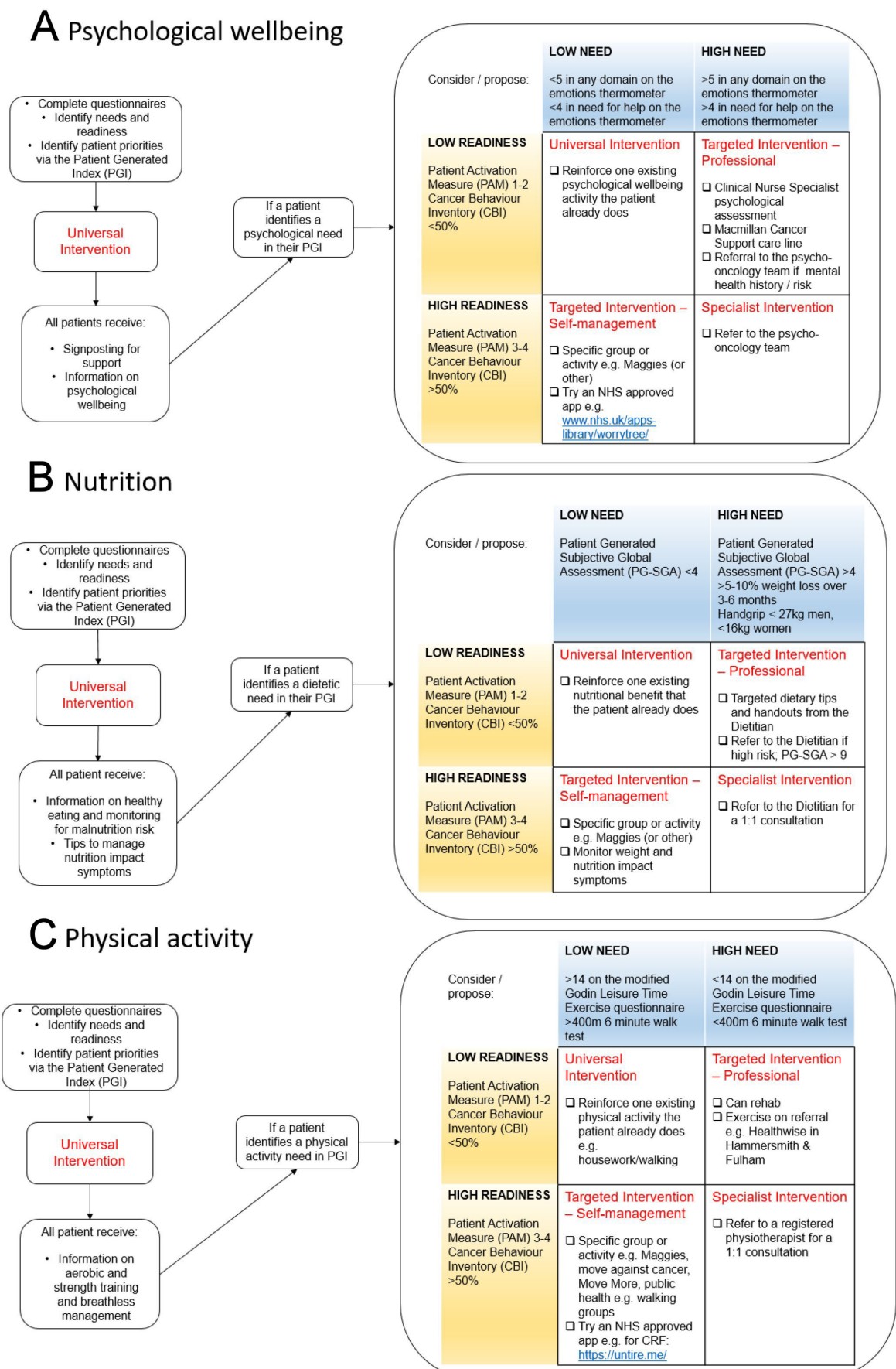

**Figure 2** (A–C) Examples of the models used in the personalised prehabilitation programme for psychological well-being, nutrition and physical activity.

**Table 1** A priori feasibility endpoints using a RAG criteria

| Parameter | Red (do not proceed to a further study) | Amber (modifications± further study required) | Green (proceed to a larger study) |
|---|---|---|---|
| Recruitment rate | <30% | 30%–60% | >60% |
| Refusal rate | >50% | 30%–50% | <30% |
| Attrition rate | >80% | 20%–80% | <20% |
| Intervention adherence | <30% | 30%–60% | >60% |

assess the impact of the personalised prehabilitation programme on:
► Overall quality of life as measured by:
  – European Organisation for Research and Treatment of Cancer (EORTC) Quality of Life Questionnaire

**Table 2** Research objectives and methods of assessment to investigate the feasibility of the personalised prehabilitation programme in clinical practice

| Research objective | Method of assessment(s) |
|---|---|
| To estimate eligibility, recruitment, refusal and attrition rates (including the reasons why) | Descriptive statistics—number approached, eligible, recruited, refused and number who drop out and at what stage<br>Field notes taken during the Lung Cancer Multi-Disciplinary Team<br>Participant discussion |
| To assess adherence to the personalised prehabilitation interventions | Weekly telephone call—adherence to goals set |
| To assess the barriers and facilitators to adherence and intervention fidelity | Daily diary—symptom log<br>Weekly telephone call<br>Short free text questionnaire at the end of the study, analysed thematically |
| To evaluate acceptability of the study processes and outcome measures | Completion rates<br>Weekly telephone call<br>Short free text questionnaire at the end of the study, analysed thematically |
| To determine the 'prehabilitation window' (length of time between diagnosis and the start of treatment) | Descriptive statistics—date of diagnosis, date of start of treatment, date of start and finish of prehabilitation |
| To determine the extent to which the intervention/s are likely to have an impact on psychological well-being, nutrition and/or physical activity and quality of life | Questionnaires—change between baseline and follow-up<br>Daily diary—symptom log<br>Weekly telephone call<br>Short free text questionnaire at the end of the study, analysed thematically<br>Functional objective assessments |
| To determine the number of participants who require universal, targeted and specialist intervention | Questionnaires—level of need and readiness<br>Descriptive statistics |

Core 30 (QLQ-C30); a cancer-specific assessment of quality of life, sensitive to changes over time.[62]
  – European Quality of Life (EuroQOL) group EQ-5D-5L for a descriptive overview of quality of life, which can be used for economic evaluation.[63]
► Functional capacity as measured by the 6MWT.[64] The 6MWT is a safe, valid and reliable measure of functional capacity in cancer populations.[65] All participants will perform a standardised, self-paced 6MWT in a 30 m long corridor. Each participant will be instructed to walk as far as they can for six minutes. Participants will be allowed to stop at any time. The distance covered (to the nearest metre) after the six minutes will be measured. A cut-off value of 400 m has been used to determine a participant's level of need for physical activity prehabilitation intervention. This is in line with a study by Kasymjanova *et al*[66] on the prognostic value of the 6MWT in advanced non-small cell lung cancer in patients undergoing chemotherapy.
► Handgrip strength as measured by a handgrip dynamometer.[67] Grip strength has shown to be correlated with overall health status.[68–72] Each participant will be instructed to sit in a chair and bend the elbow of their non-dominant arm at a 90° angle. The lead researcher will ask the participant to squeeze the device as hard as they can. Each participant will repeat this three times and the average will be recorded. A cut-off value of 27 kg for men and 16 kg for women has been used to determine a participant's level of need for nutritional prehabilitation intervention. This is based on the revised European consensus guidelines on the definition and diagnosis of sarcopenia.[73]

To tailor the intervention, the above outcome measures will be assessed prior to starting treatment and before the personalised prehabilitation programme begins (baseline), at week three and at week six of a participant's oncological treatment (end point).

Demographic and clinical data including age, gender, level of education achieved, ethnicity, employment status, level of social support (measured using the Duke-University of North Carolina Functional Social Support Questionnaire),[74] diagnosis, staging, treatment and comorbidities (measured using the Adult Co-morbidity Evaluation-27 score)[75] will be collected at baseline.

### Training
The lead researcher is a highly specialised oncology dietitian trained in advanced communication and psychological skills[76] to manage complex conversations with patients and help recognise and respond to psychological distress. The lead researcher is familiar with using a handgrip dynamometer from previous experience in clinical practice. Training in conducting the 6MWT has been provided by a specialist oncology physiotherapist.

### Patient and public involvement
Four patients (and a spouse) who had recently completed chemotherapy or radiotherapy treatment for

lung cancer were consulted on the personalised prehabilitation programme via a focus group. This was separate to focus groups used in the initial codesign process to develop the personalised prehabilitation programme, which used a three-step participatory action research approach (see online supplemental material file 1).[36–38] Members of the focus group were very positive about the programme, with many patients reporting that they found having a strong mind-set, staying active and eating well helped them get through treatment. These are key components of the personalised prehabilitation programme. The focus group also reviewed the study questionnaires and patient documentation. Patients and their relatives expressed the benefits of having a weekly telephone call to discuss their concerns and reported that the questionnaires and daily diary would make them feel supported, as the information would be tailored to them individually.

All participants who participate in the study will be offered the opportunity to provide feedback on their experience of the personalised prehabilitation programme via the use of a short free text questionnaire. This feedback will help inform future programme adjustments and have the potential to improve patient experience.

### Data collection and management
Data will be collected and stored according to the Data Protection Act (2018) and in line with General Data Protection Regulation (GDPR).

All data included in this study will be kept as part of the participants clinical care record on a secure password-protected electronic patient record system at the study hospital.

A subset of this data will be extracted and entered into a case report form, which will be kept on a secure password-protected database at the study hospital and only accessed by the research team. This data will be pseudonymised and all participants will be given a unique study number.

### Statistical analysis plan
This study includes a small cohort of lung cancer participants from a single London teaching hospital. There are no groups to detect differences between, therefore, no formal tests of statistical significance will be undertaken.

Descriptive statistics will be used to analyse baseline demographic and clinical data. Continuous measures will be reported as means, standard deviations and/or medians along with confidence intervals, while categorical data will be reported as percentages. Missing data will be reported where applicable.

Inductive thematic analysis will be used to analyse qualitative data from the short free text questionnaire regarding patient experience.

Feasibility will be determined if the a priori thresholds for feasibility have been met.

The data collected will be used to inform the design of a larger randomised pilot study.

### Strengths and limitations
This study uses a novel way of delivering a multimodal prehabilitation programme through the use of a personalised, inclusive research approach that aligns with patient values, needs and functional goals, with interventions tailored to a participant's capabilities. The personalised prehabilitation programme provides a structure and process for personalising prehabilitation that is person-centred and underpinned by a conceptual framework, which is consistent with psychological adjustment theory. This could be key to empower patients to maintain progress through the longer, more variable context of perioncological prehabilitation.

This study will help address key uncertainties, which need to be resolved before implementation, including engagement and adherence to prehabilitation interventions in patients with lung cancer receiving chemotherapy, radiotherapy or immunotherapy treatment.

This study is limited by (A) lack of a comparison group, (B) small sample size, (C) self-reported adherence to the prehabilitation interventions and (D) the lead researcher also being the clinician delivering the interventions and performing assessments. In order to mitigate potential bias regarding the latter, where possible, aspects of the programme have been predefined.

### ETHICS AND DISSEMINATION
Ethical approval was obtained from the Health Research Authority (HRA) on 13 April 2022 (reference number 22/PR/0390) prior to the start of the study. Any modifications will be approved by the HRA. This study has been registered with ClinicalTrials.gov (NCT: NCT05318807). Imperial College Healthcare NHS Trust is the sponsor of the study. Monitoring and auditing will be conducted in accordance with the sponsor's policies and procedures. Results of this study will be disseminated to academics, stakeholders, cancer alliances, relevant charities and to the public and people with cancer. This will be through publication in peer-reviewed articles, presentations at scientific conferences, social media and collaboration with patient and public involvement representatives.

**Contributors** KW-M is the principal investigator. KW-M devised the clinical question, applied for funding and research ethics and was involved in the conception and design of the study and protocol. AK contributed to the study design and methods. AK and CU were involved in the conception and design of the study. LJ and MW contributed to all versions of the protocol. AK, CU and MW provided their expertise in cancer care and contributed to the knowledge needed for prehabilitation. CU, LJ and MW provided supervision and mentorship. All authors read and approved the final protocol.

**Funding** This work was supported by Health Education England and the National Institute for Health and Care Research as part of an ICA Pre-doctoral Clinical and Practitioner Academic Fellowship (Grant award number: NIHR302680).

**Competing interests** None declared.

**Patient and public involvement** Patients and/or the public were involved in the design, or conduct, or reporting, or dissemination plans of this research. Refer to the Methods section for further details.

**ORCID iDs**
Kelly Wade-Mcbane http://orcid.org/0000-0001-6509-0359
Alex King http://orcid.org/0000-0001-9050-1092
Catherine Urch http://orcid.org/0000-0002-4251-9113
Lina Johansson http://orcid.org/0000-0002-8571-0567
Mary Wells http://orcid.org/0000-0001-5789-2773

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
