## [Reviewer comments · BMJ Open]

ARTICLE DETAILS

TITLE (PROVISIONAL)	Is personalised prehabilitation feasible to implement for patients undergoing oncological treatment for lung cancer at a London teaching hospital? Protocol of a feasibility trial.
AUTHORS	Wade-Mcbane, Kelly; King, Alex; Urch, Catherine; Johansson, Lina; Wells, Mary

VERSION 1 – REVIEW

REVIEWER	Chevalier, Stéphanie McGill University, School of Human Nutrition
REVIEW RETURNED	17-Mar-2023

GENERAL COMMENTS	General comments : This paper describes the protocol of a feasibility study on a stratified prehabilitation intervention in patients with lung cancer undergoing non-surgical treatment. This trial is relevant considering the high proportion of patients with lung cancer diagnosed in late stages with the need for non-surgical treatment in addition to the poor prognosis of the disease. The patient-centered intervention has many strengths including the stratification of the intervention allowing for better resource management, and that is has been co-created with patient partners. However, the interventions (nutrition and exercise) are not sufficiently described and detailed for reproducibility. The following are points that would require clarification. Specific comments : Introduction: Although not in lung cancer patients, prehabilitation has been performed in patients undergoing chemotherapy in rectal cancer and results from previous studies could be presented in the introduction to support the current trial. For instance, PMID: 25274049. Methods Page 3: In addition to the feasibility study framework cited, it is suggested to follow the guidelines of CONSORT 2010 statement: extension to randomized pilot and feasibility trials (BMJ 2016; 355 doi: https://doi.org/10.1136/bmj.i5239). Although this not a randomized trial, many of the checklist points would apply and merit to be verified to comply to best research practises and reporting. Page 3, line 30: please add: this open-label, single-group study... Page 3, line 39: Eligibility criteria: do they include all lung cancer types and stages, and the presence of any comorbidity and medication used? Page 4, lines 20-30: though indicated on Figure 3, cut-offs for each questionnaire should be indicated in the text.
---

	Page 4, line 51: It is unclear what the 1:1 sessions will consist of for specialist interventions. Will the sessions with the physiotherapist serve to increase physical activity? If so, will you be focusing on resistance or aerobic training? Same comments apply for sessions with the dietitian. What will you be assessing, how, and what will be the objective of your intervention? Figure 3: There are 2 questionnaires/criteria for each readiness and need assessment category. How are these computed; do both criteria need to be present to meet the need, or only one of the two? What if there are conflicting results between two tests to classify participants? E.g. Godin Leisure Time Exercise score > 14 but 6MWT < 400m? What if both PG-SGA and grip strength are low? Page 4, line 49-50: Revise sentence, has no verb. Page 5, line 3-4: It is not clear from the text nor the Figure 3 how the 6-minute walk test and the grip strength measures will help to tailor the intervention. Please clarify. Page 5, Table 1: Given the number of research objectives and the expected heterogeneity of participant characteristics, it is questionable whether the sample size of 30 will be sufficient, especially for the qualitative assessments (e.g. who is suitable, barriers and facilitators). It may prove useful to limit the number of primary objectives. As well, the 4th objective appears very similar to the first. Table 1, Objective 2: Will it be possible to perform field notes and use data from participant discussions in patients who do not participate in the program? Would they need to provide consent for this? Page 6, line 30: The sentence "The benefits of prehabilitation can be seen in as little as two weeks" could be clarified as to which specific benefits can be detected (functional, psychological, ...). Page 6, Secondary objectives and outcome measures section: please indicate the justification for choosing the 400 m cut-off for 6MWT and the 27/16 kg for handgrip strength to determine the need to receive a physical/nutritional prehabilitation intervention. Page 7, Statistical Analysis: How will the study be determined as feasible or not, based on all primary outcome measures or a few selected one? Importantly, the thresholds for feasibility criteria (recruitment rate, attrition, adherence, ...) should be determined a priori and stated. Also, please report confidence intervals in descriptive data. Page 8, Strengths and limitations: please add "the lead researcher is also the clinician delivering the intervention and performing assessments", which can introduce bias. An additional limitation is the self-reported adherence as there are no objective measures of physical activity or nutrition. Page 8, line 29: indicate the NCT registration number on ClinicalTrials.gov. The title registered mentions patients having chemotherapy or radiotherapy, but not immunotherapy, as in the manuscript. Make sure to harmonize both protocols. Figure 1: this text could be simply stated as a 3-step approach in the method section. There is no visual benefit to present this text in a figure. Figure 2: the visuals are not self-explanatory, could be clarified? Do the size of shapes mean anything?
--	--

REVIEWER	Bayly, Joanne King's College London, Cicely Saunders Institute for Palliative Care, Policy and Rehabilitation
REVIEW RETURNED	31-Mar-2023

GENERAL COMMENTS	General comments: It was a pleasure to read this manuscript. It is encouraging to see another study investigating non-exercise based, multi-modal personalised rehabilitation in people with lung cancer at the start of treatment where research is scarce. This population is at high risk for significant disability prior to diagnosis and through treatment. The theoretical rationale and approach are robust. Previous studies of non-exercise-based rehabilitation interventions in this population have also been feasibility designs and so numbers are small. I look forward to reading the findings in due course. Introduction: Pg 2 Lines 31-37 The authors haven't cited the intervention development work reported by this reviewer (Bayly et al 2018 for the feasibility RCT reported by Bayly et al in 2019. Both studies were conducted in a comparable population and tested a comparable model of care. They provide additional support the model of care and approach taken in this study. Aims- Can you clarify if you intend the findings to support implementation of the intervention into clinical practice, or to support a future definitive trial? Methods and Strengths and Limitations. Did you use a reporting guideline to structure this protocol and if not, why not. Study design- can you provide more detail on your research design (it appears to be a mixed methods study- what mixed methods design are you utilizing? Have you situated it within the updated MRC framework for the development and testing of complex interventions? This will strengthen the methods section within the protocol. Please report more detail on your co-design work (which is a strength). For example, how were the six patients from the single London Teaching hospital recruited (possible limitation due to potential bias in the focus groups). Did anyone decline? Were those who declined different in anyone from those who agreed to participate? Were there any divergent views on the proposed model of care and study design?
---

VERSION 1 – AUTHOR RESPONSE

Reviewer 1 comments

- 1. Introduction: Although not in lung cancer patients, prehabilitation has been performed in patients undergoing chemotherapy in rectal cancer and results from previous studies could be presented in the introduction to support the current trial. For instance, PMID: 25274049.**

Response: Thank you for this suggestion. Accordingly, we have revised the introduction to include the following: 'A small number of studies have investigated the use of prehabilitation in the neo-adjuvant setting and have shown benefit in terms of post-surgical outcomes, but only one of these have been in lung cancer.' Please see page 2, lines 8 – 11 of the introduction.

- 2. Methods - Page 3: In addition to the feasibility study framework cited, it is suggested to follow the guidelines of CONSORT 2010 statement: extension to randomized pilot and**

feasibility trials (BMJ 2016; 355 doi: <https://doi.org/10.1136/bmj.i5239>). Although this not a randomized trial, many of the checklist points would apply and merit to be verified to comply to best research practises and reporting.

Response: Thank you for this suggestion. Please find attached a completed copy of the CONSORT 2010 statement: extension to randomized pilot and feasibility trials checklist, titled 'Supplemental material. File 1. CONSORT checklist.

3. Page 3, line 30: please add: this open-label, single-group study...

Response: Thank you for this suggestion. Accordingly, we have revised the sentence regarding our study design as follows: 'Using the Orsmond & Cohn framework for feasibility studies, this open-label, single-group study will use a mix of methods to understand the different aspects of feasibility.' Please see page 3, line 5 of methods and analysis.

4. Page 3, line 39: Eligibility criteria: do they include all lung cancer types and stages, and the presence of any comorbidity and medication used?

Response: Thank you for this comment. Accordingly, we have revised the wording of our eligibility criteria as follows: Adults (≥ 18 years) with a diagnosis of lung cancer who are due to commence chemotherapy, radiotherapy or immunotherapy treatment. This study includes all lung cancer types and stages. Patients with co-morbidities and those on medication will be considered eligible. Interventions will be tailored to a participant's capabilities, thereby promoting inclusivity and optimisation of participant recruitment. Please see page 3.

5. Page 4, lines 20-30: though indicated on Figure 3, cut-offs for each questionnaire should be indicated in the text.

Response: Thank you for this comment. Cut-offs for each questionnaire have now been included in the text as follows:

- *Cancer Behaviour Inventory (CBI) brief form - to measure self-efficacy for coping with cancer. A score of less than 50% indicates a low level of readiness, whereas a score of 50% or greater indicates a high level of readiness.*
- *Patient Activation Measure (PAM) - to measure a person's knowledge, skills and confidence in managing their own health and wellbeing. A score of one-two indicates a low level of readiness, whereas a score of three-four indicates a high level of readiness.*
- *Modified Godin leisure time exercise questionnaire - to measure activity. A score of 14 or greater demonstrates that a person is active, thereby indicating a low level of need for physical activity. Conversely, a score less than 14 demonstrates that a person is insufficiently active, thereby indicating a high level of need to optimise activity levels.*
- *Patient Generated Subjective Global Assessment (PG-SGA) short form - to identify malnutrition risk. A score less than four demonstrates no or minimal risk for malnutrition, thereby indicating a low level of need for nutritional support. Conversely, a score of four or greater demonstrates that a person is at risk of malnutrition or is malnourished, thereby indicating a high level of need for nutrition support.*
- *Emotions Thermometer (ET) - to evaluate levels of emotional distress. A score in any domain less than five or a score of less than four in need for help demonstrates low levels of emotional distress, thereby indicating a low level of need. Conversely, a score of five or greater in any domain or a score of four or greater in need for help demonstrates high levels of emotional distress, thereby indicating a high level of need.*
- *Patient Generated Index (PGI) - to identify participant priorities.*

Please see page 4.

Please note: Owing to the removal of figure 1, figure 3 is now referred to as figure 2 in this revised document.

- 6. Page 4, line 51: It is unclear what the 1:1 sessions will consist of for specialist interventions. Will the sessions with the physiotherapist serve to increase physical activity? If so, will you be focusing on resistance or aerobic training? Same comments apply for sessions with the dietitian. What will you be assessing, how, and what will be the objective of your intervention?**

Response: Thank you for this comment and apologies for the lack of clarity. Accordingly, we have revised this section to clarify what we will be assessing, how we will be assessing it and what the objective of the intervention will be as follows:

'The personalised prehabilitation programme uses a universal, targeted and specialist approach to prehabilitation interventions, in line with current guidance. All participants will receive a universal intervention comprising of written information and relevant signposting to cancer specific nutrition, physical activity and/or psychosocial wellbeing support resources. If a participant identifies a psychological, nutrition and/or physical activity need, they will then receive the appropriate intervention according to their level of need and readiness (see figures 2a, 2b and 2c). Targeted interventions will be delivered via the use of certified apps and/or specific group activities at Maggie's cancer centre (an independent charity that provides free cancer support, workshops and courses to help anyone affected by cancer). For example, an exercise class for participants undergoing treatment, nutritional education sessions as well as a lung cancer support group. These group activities are run by cancer support workers or qualified instructors. Specialist interventions involve a 1:1 assessment with a registered healthcare professional such as a physiotherapist, dietitian and/or a clinical psychologist depending on a participant's needs. The objectives of specialist intervention with the physiotherapist and dietitian are to increase physical activity and optimise dietary intake to minimise loss of body weight and muscle mass, respectively. Specialist intervention with the physiotherapist will include tailored advice regarding resistance and aerobic exercise. In comparison, specialist intervention with the dietitian will include a full individualised nutritional assessment, taking into account anthropometric, biochemical and clinical parameters as well as undertaking a comprehensive diet history and calculation of nutritional requirements to minimise further malnutrition.' Please see page 5, lines 61-70 of the personalised prehabilitation programme.

- 7. Figure 3: There are 2 questionnaires/criteria for each readiness and need assessment category. How are these computed; do both criteria need to be present to meet the need, or only one of the two? What if there are conflicting results between two tests to classify participants? E.g. Godin Leisure Time Exercise score > 14 but 6MWT < 400m? What if both PG-SGA and grip strength are low?**

Response: Thank you for this comment. To improve clarity regarding the criteria used for the assessment of readiness and need, we have included the following in the text: 'Questionnaire responses and outcomes from the functional measurements will inform the conversation with the participant about their priorities and enable interventions to be stratified according to a participant's level of readiness and need. If a subjective (questionnaire) or objective measurement (6MWT or handgrip strength) indicates a low or high level of readiness or need, the intervention will be tailored to the individual accordingly. See figures 2a, 2b and 2c.' Please see page 5, lines 43-48 of the personalised prehabilitation programme.

Please note: Owing to the removal of figure 1, figure 3 is now referred to as figure 2 in this revised document.

- 8. Page 4, line 49-50: Revise sentence, has no verb.**

Response: Thank you for this comment. Accordingly, we have revised the sentence to include a verb, as follows: 'For example, exercise classes or nutritional education sessions for participants undergoing treatment as well as the opportunity to attend a lung cancer support group.' Please see page 5, lines 59-60 of the personalised prehabilitation programme.

9. Page 5, line 3-4: It is not clear from the text nor the Figure 3 how the 6-minute walk test and the grip strength measures will help to tailor the intervention. Please clarify.

Response: Thank you for this comment. We hope that we have addressed this comment in our response to comment 15 above. Please note: Owing to the removal of figure 1, figure 3 is now referred to as figure 2 in this revised document.

10. Page 5, Table 1: Given the number of research objectives and the expected heterogeneity of participant characteristics, it is questionable whether the sample size of 30 will be sufficient, especially for the qualitative assessments (e.g. who is suitable, barriers and facilitators). It may prove useful to limit the number of primary objectives. As well, the 4th objective appears very similar to the first.

Response: Thank you for this comment. We agree with the reviewer that there is likely to be expected heterogeneity of participant characteristics. Accordingly, we have acknowledged this in the manuscript as follows: 'Heterogeneity of participant characteristics is expected within the study. This will provide valuable insights into how different people experience the personalised prehabilitation programme.' Please see page 7, line 12 of primary objective and outcome measure.

The research objectives have also been revised and reduced from 11 to 7. We feel these will be achievable within the time frame of the feasibility study. Please see page 6.

Please note: Table 1 is now referred to as table 2 in this revised document.

11. Table 1, Objective 2: Will it be possible to perform field notes and use data from participant discussions in patients who do not participate in the program? Would they need to provide consent for this?

Response: Thank you for this comment. Ethical approval from the Health Research Authority (reference number 22/PR/0390) has been obtained for the lead researcher to keep field notes when attending the lung cancer multi-disciplinary team (MDT) meeting to observe how the MDT determines who should and who should not participate in the prehabilitation programme.

Please note: Table 1 is now referred to as table 2 in this revised document.

12. Page 6, line 30: The sentence "The benefits of prehabilitation can be seen in as little as two weeks" could be clarified as to which specific benefits can be detected (functional, psychological, ...).

Response: Thank you for this comment. Accordingly, we have revised the sentence as follows: 'Functional benefits of prehabilitation can be seen in as little as two weeks.' Please see page 7, line 2 of secondary objectives and outcome measures.

13. Page 6, Secondary objectives and outcome measures section: please indicate the justification for choosing the 400 m cut-off for 6MWT and the 27/16 kg for handgrip strength to determine the need to receive a physical/nutritional prehabilitation intervention.

Response: Thank you for this comment. Accordingly, we have revised this section to provide a justification for choosing the 400m cut off for the 6MWT as follows: 'A cut off value of 400m has

been used to determine a participant's level of need for physical activity prehabilitation intervention. This is in line with a study by Kasymjanova et al on the prognostic value of the 6MWT in advanced non-small cell lung cancer in patients undergoing chemotherapy. Please see page 7, lines 15-18 of secondary objectives and outcome measures.

Similarly, we have also provided a justification for choosing the following cut off values for handgrip strength: 'A cut off value of 27kg for men and 16kg for women has been used to determine a participant's level of need for nutritional prehabilitation intervention. This is based on the revised European consensus guidelines on the definition and diagnosis of sarcopenia.' Please see page 7, lines 23-27 of secondary objectives and outcome measures.

14. Page 7, Statistical Analysis: How will the study be determined as feasible or not, based on all primary outcome measures or a few selected one? Importantly, the thresholds for feasibility criteria (recruitment rate, attrition, adherence, ...) should be determined a priori and stated. Also, please report confidence intervals in descriptive data.

Response: Thank you for these comments and suggestions. Accordingly, we have included a priori feasibility endpoints using a RAG criteria, which are outlined in table 1. Please see page 6. We have included a sentence on how the study will be determined as feasible as follows:

'Feasibility will be determined if the a priori thresholds for feasibility have been met.' Please see page 9, line 11 of statistical analysis plan.

We have also included confidence intervals in our statistical analysis, which is reported as follows: 'Continuous measures will be reported as means, standard deviations and/or medians along with confidence intervals, while categorical data will be reported as percentages. Missing data will be reported where applicable.' Please see page 9, lines 6-8 of statistical analysis plan.

15. Page 8, Strengths and limitations: please add "the lead researcher is also the clinician delivering the intervention and performing assessments", which can introduce bias. An additional limitation is the self-reported adherence as there are no objective measures of physical activity or nutrition.

Response: Thank you for these comments. Accordingly, we have revised our sentence that the lead researcher is also the clinician delivering the interventions to also include performing assessments as follows: 'The lead researcher also being the clinician delivering the interventions and performing assessments.' Please see page 9, lines 13-14 of strengths and limitations.

We have also added the following to the strengths and limitations section: 'Self-reported adherence to the prehabilitation interventions.' Please see page 9, line 13 of strengths and limitations.

16. Page 8, line 29: indicate the NCT registration number on ClinicalTrials.gov. The title registered mentions patients having chemotherapy or radiotherapy, but not immunotherapy, as in the manuscript. Make sure to harmonize both protocols.

Response: Thank you for bringing this to our attention and apologies for our oversight. We have now included the NCT registration number on ClinicalTrials.gov; 22CX7570. Please see page 9, line 5 of ethics and dissemination. We have also updated the trial registry accordingly to ensure that it is consistent with the protocol manuscript.

17. Figure 1: this text could be simply stated as a 3-step approach in the method section. There is no visual benefit to present this text in a figure.

Response: Thank you for this comment. Accordingly, we have revised the PPI section, as follows: 'This was separate to focus groups used in the initial co-design process to develop the personalised prehabilitation programme, which used a three-step approach. See supplemental

material file 2.' Please see page 8, line 5 of patient and public involvement. Please note: We have removed figure 1 and we have provided further information about the co-design process as supplementary material, titled 'Supplemental material. File 2. Co-design process.

18. Figure 2: the visuals are not self-explanatory, could be clarified? Do the size of shapes mean anything?

Response: Thank you for this comment. Owing to the removal of figure 1, figure 2 is now referred to as figure 1 in this revised document. The description of this figure has been revised to improve clarity as follows: 'Figure one. Diagrammatic representation of the Adversity, Restoration and Compatibility (ARC) model of the experience of adjusting to living with and beyond cancer. The waves and curves reflect the fluidity and non-linear nature of process, with the sizes reducing as one adjusts to living with and beyond cancer.'

Reviewer 2 comments

19. Introduction - Pg 2 Lines 31-37 The authors haven't cited the intervention development work reported by this reviewer (Bayly et al 2018 for the feasibility RCT reported by Bayly et al in 2019. Both studies were conducted in a comparable population and tested a comparable model of care. They provide additional support the model of care and approach taken in this study.

Response: Thank you for bringing this work to our attention. Accordingly, we have revised our introduction to include the work of this reviewer as follows: 'This is further supported by Bayly et al's research in thoracic lung cancer, which found that participants had varied needs, highlighting the importance of tailored support to optimise symptom control, treatment tolerance and independence. Please see page 1, lines 26-29 of the introduction.

20. Aims - Can you clarify if you intend the findings to support implementation of the intervention into clinical practice, or to support a future definitive trial?

Response: Thank you for this comment. We intend the findings to support a future definitive study and thus, to improve clarity, we have amended the following sentence to reflect this: 'This proposed feasibility study aims to investigate whether a personalised prehabilitation programme co-designed with healthcare professionals and patients and informed by theory and evidence is feasible to implement and evaluate in clinical practice and could thereby support a future definitive study.' Please see page 3, lines 56-57 of the introduction.

21. Methods and Strengths and Limitations - Did you use a reporting guideline to structure this protocol and if not, why not.

Response: Thank you for this comment and apologies for our oversight in our documentation of this. We used The Consolidated Standards of Reporting Trials (CONSORT) reporting guideline for pilot and feasibility studies and we have provided a completed checklist for this, titled 'Supplemental material. File 1. CONSORT checklist.' Please see page 3, lines 13-15 of methods and analysis.

22. Study design- can you provide more detail on your research design (it appears to be a mixed methods study- what mixed methods design are you utilizing?)

Response: Thank you for this comment. We will be using a mix of methods to understand the different aspects of feasibility and we have added this to our study design in the manuscript. Please see page 3, lines 5-6 of methods and analysis.

23. Have you situated it within the updated MRC framework for the development and testing of complex interventions? This will strengthen the methods section within the protocol.

Response: Thank you for this comment. This protocol will be situated within the feasibility stage of the MRC framework for developing and evaluating complex interventions and this has now been documented in the manuscript. Please see page 3, lines 9-11 of methods and analysis.

24. Please report more detail on your co-design work (which is a strength). For example, how were the six patients from the single London Teaching hospital recruited (possible limitation due to potential bias in the focus groups). Did anyone decline? Were those who declined different in anyone from those who agreed to participate? Were there any divergent views on the proposed model of care and study design?

Response: Thank you for this comment. More information about the co-design work has been provided as supplementary material, titled 'Supplemental material. File 2. Co-design process.'

VERSION 2 – REVIEW

REVIEWER	Bayly, Joanne King's College London, Cicely Saunders Institute for Palliative Care, Policy and Rehabilitation
REVIEW RETURNED	16-May-2023

GENERAL COMMENTS	The manuscript is much improved. The qualitative focus groups with patients and clinicians are better described, but I'm not sure that researchers in this field of EBCD will be satisfied with the level of detail provided in the supplementary material. There are no citations provided for the EBCD methods. I recommend describing this element of the work as qualitative group interviews to develop the intervention guided by the MRC framework, rather than EBCD. It could be more concise - for example you could cite the standard method to collect 6MWT and hand grip test data rather than describe in the paper the details, with the cut of details remaining in the text.
--

VERSION 2 – AUTHOR RESPONSE

Reviewer 2 comments

- 1. The qualitative focus groups with patients and clinicians are better described, but I'm not sure that researchers in this field of EBCD will be satisfied with the level of detail provided in the supplementary material. There are no citations provided for the EBCD methods. I recommend describing this element of the work as qualitative group interviews to develop the intervention guided by the MRC framework, rather than EBCD**

Response: Thank you for this suggestion. Accordingly, we have revised our co-design process supplemental material document. We have also added a second cross-reference to this document earlier in the manuscript. Please see page 2, line 44. Owing to this, Supplemental material. File 2. Co-design process is now referred to as Supplemental material. File 1. Co-design process. The previous supplemental material. File 1. CONSORT checklist is now referred to as Supplemental material. File 2. CONSORT checklist. These changes have been reflected in the main document and the updated numbered supplemental files are attached.

- 2. It could be more concise - for example you could cite the standard method to collect 6MWT and hand grip test data rather than describe in the paper the details, with the cut of details remaining in the text.**

Response: Thank you for this comment. We feel that a detailed description of these measures is helpful to readers who may question whether patients with lung cancer are 'fit' enough to participate in exercise, as it shows this is being considered very carefully. We would like this text to remain as we believe it increases the transparency of our paper. A similar level of detail has also been observed in other protocol papers within the field of cancer prehabilitation.

VERSION 3 – REVIEW

REVIEWER	Bayly, Joanne King's College London, Cicely Saunders Institute for Palliative Care, Policy and Rehabilitation
REVIEW RETURNED	29-Jun-2023
GENERAL COMMENTS	I've recommended that the protocol is accepted for publication and look forward to reading the results in due course. I've enjoyed reading the protocol and am particularly impressed with the application of theory to the intervention and your clear description of the intervention and measurements. It may be worth when you publish the results, letting readers know if there are any cost implications for using any of the measures, for example the PAM. I know that this was funded in some areas of the NHS but am not sure if this is still the case, or if it is the case in other countries. I would have liked a citation for the co-design methodology- but it's a minor point.

VERSION 3 – AUTHOR RESPONSE

Reviewer 2 comments

- 1. I've recommended that the protocol is accepted for publication and look forward to reading the results in due course. I've enjoyed reading the protocol and am particularly impressed with the application of theory to the intervention and your clear description of the intervention and measurements. It may be worth when you publish the results, letting readers know if there are any cost implications for using any of the measures, for example the PAM. I know that this was funded in some areas of the NHS but am not sure if this is still the case, or if it is the case in other countries. I would have liked a citation for the co-design methodology- but it's a minor point.**

Response: Thank you for your comments. We will take into consideration your suggestion about informing readers of any cost implications for using any of the measures when we publish the results. We have also included citations for the co-design methodology. Please see page 2, lines 43-45 and page 8, lines 17-19.